# Nonlinear plasmon-exciton coupling enhances sum-frequency generation from a hybrid metal/semiconductor nanostructure

Jin-Hui Zhong [1,7], Jan Vogelsang [2,3,7], Jue-Min Yi[1], Dong Wang [4], Lukas Wittenbecher[2,3], Sara Mikaelsson[2], Anke Korte[1], Abbas Chimeh[1], Cord L. Arnold[2], Peter Schaaf [4], Erich Runge [5], Anne L' Huillier[2], Anders Mikkelsen[2,3] & Christoph Lienau [1,6✉]

The integration of metallic plasmonic nanoantennas with quantum emitters can dramatically enhance coherent harmonic generation, often resulting from the coupling of fundamental plasmonic fields to higher-energy, electronic or excitonic transitions of quantum emitters. The ultrafast optical dynamics of such hybrid plasmon–emitter systems have rarely been explored. Here, we study those dynamics by interferometrically probing nonlinear optical emission from individual porous gold nanosponges infiltrated with zinc oxide (ZnO) emitters. Few-femtosecond time-resolved photoelectron emission microscopy reveals multiple long-lived localized plasmonic hot spot modes, at the surface of the randomly disordered nanosponges, that are resonant in a broad spectral range. The locally enhanced plasmonic near-field couples to the ZnO excitons, enhancing sum-frequency generation from individual hot spots and boosting resonant excitonic emission. The quantum pathways of the coupling are uncovered from a two-dimensional spectrum correlating fundamental plasmonic excitations to nonlinearly driven excitonic emissions. Our results offer new opportunities for enhancing and coherently controlling optical nonlinearities by exploiting nonlinear plasmon-quantum emitter coupling.

---

[1] Institute of Physics, Carl von Ossietzky University, 26111 Oldenburg, Germany. [2] Department of Physics, Lund University, SE-221 00 Lund, Sweden. [3] Nano Lund, Lund University, Box 118, 22100 Lund, Sweden. [4] Institut für Mikro- und Nanotechnologien MacroNano® and Institut für Werkstofftechnik, Technische Universität Ilmenau, 98693 Ilmenau, Germany. [5] Institut für Mikro- und Nanotechnologien MacroNano® and Institut für Physik, Technische Universität Ilmenau, 98693 Ilmenau, Germany. [6] Forschungszentrum Neurosensorik, Carl von Ossietzky University, 26111 Oldenburg, Germany. [7] These authors contributed equally: Jin-Hui Zhong, Jan Vogelsang. ✉email: christoph.lienau@uni-oldenburg.de

Nonlinear plasmonics of metallic nanoantennas is highly promising for creating subwavelength coherent light[1] and electron[2] sources, and for ultrafast all-optical control[3]. Importantly, the geometry of such nanoantennas can be flexibly designed, allowing to tune their plasmonic modes to be resonant with the fundamental or higher harmonic frequencies of the driving laser[4] or even to be doubly resonant[5,6]. The nonlinear efficiency of pure plasmonic nanoantennas, however, is limited by their intrinsically weak optical nonlinearities and the weak penetration of light into the metal. To further enhance nonlinear emission, hybrid metal–semiconductor nanostructures have been fabricated by combining the strong field enhancements of plasmons and the large nonlinear susceptibility of semiconductors[7–14]. The enhancement of nonlinear emission often results from the coupling of fundamental plasmonic fields to higher-energy, electronic[7] or excitonic[9,14,15] transitions of quantum emitters. Although many studies have been performed on various combinations of plasmonic nanoantennas and semiconductors in the frequency domain, considerably less is known about their ultrafast optical dynamics. Accessing such dynamics is not only important to understand the microscopic origin of the coupling and the mechanism of nonlinear signal enhancement, but also necessary for full coherent control of coupled nanosystems[3]. Such femtosecond dynamics have been probed on pure plasmonic nanostructures and have proven crucial in understanding plasmonic coherence properties[16–19]. Furthermore, most studies on the hybrid nanostructures focus on enhancing second- or third-harmonic generation that doubles or triples energetically degenerate photons, respectively. Other nonlinear wave-mixing channels, such as sum-frequency (SF) generation, that provide more possibilities and tunabilities to mix different colors of fundamental photons, have been rarely explored.

Here, by using few-cycle driving pulses, we record interferometric time-resolved electron and light emission from bare Au and hybrid Au/ZnO nanosponges. We show that the coupling between plasmonic hot spot fields and excitonic quantum emitters, via SF quantum channels, boosts nonlinear excitonic emission from ZnO. The results are valuable for improving and controlling optical nonlinearity via nonlinear plasmon–quantum emitter coupling[20].

## Results

**Nonlinear light emission from localized plasmonic hot spots.** For our studies of nonlinear plasmon–exciton coupling, we take gold nanosponges—porous half-spherical nanoparticles with diameters of several hundreds of nanometers—as a promising plasmonic nanoantenna[21]. Such nanosponges are perforated with 10–20 nm sized nanopores throughout the entire particle, forming a randomly disordered ligament network (see scanning electron microscopy (SEM) image in Fig. 1a, more SEM images can be found in Supplementary Fig. 1). Their unique structure allows us not only to efficiently couple far-field light to the dipolar mode of the particle, but also to funnel the electromagnetic energy into a series of localized hot spots via multiple coherent scattering of surface plasmons in the disordered structure[22,23]. Dark-field white light scattering spectra show a broad resonance superimposed with some random spectral modulations, reflecting the excitation of multiple hot spot modes superimposed to the dipole mode of single nanosponges (Supplementary Fig. 3). Since

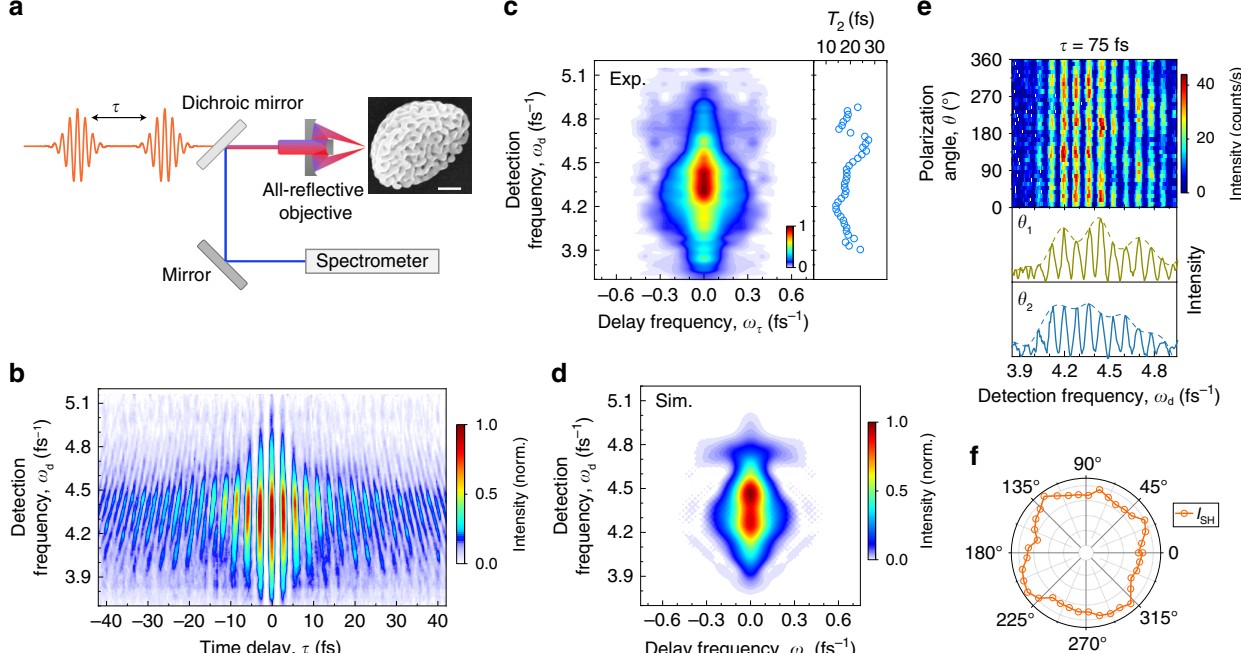

**Fig. 1 Time-resolved nonlinear light emission from long-lived localized plasmonic hot spots in a gold nanosponge. a** Scheme of the interferometric frequency-resolved autocorrelation (IFRAC) experiment. A broadband, phase-locked 8-fs pulse pair is used to excite and time resolve the plasmonic fields of a single gold nanosponge—a porous half-spherical nanoparticle with a randomly disordered ligament network (see scanning electron microscopy image, scale bar: 100 nm). The back-scattered nonlinear signal is detected in a spectrometer as a function of inter-pulse delay $\tau$. **b** IFRAC trace of a single gold nanosponge. **c** Zeroth-order (DC) band of the Fourier transform of the IFRAC trace along $\tau$. The clear stretching along the detection axis $\omega_d$ indicates an inhomogeneous broadening of the plasmonic hot spots of the nanosponge. Right inset: fitted homogeneous dephasing time of hot spots of ~16 fs as a function of detection frequency. **d** Simulated DC band by taking the incoherent sum of the second-harmonic (SH) emission from five hot spots, see details in Supplementary Fig. 13. **e** Top: excitation-polarization-resolved SH spectra at a fixed $\tau \sim 75$ fs; bottom: representative SH spectra taken at $\theta_1 = 175°$ and $\theta_2 = 320°$, showing broadband, multi-peaked spectral structures. **f** Polar plot of the integrated SH intensity. A multipole feature is characteristic for SH emission from multiple randomly oriented hot spots.

the scattering spectra are largely inhomogeneously broadened, they do not give direct insight into the plasmonic properties of single hot spots. Light scattering from individual hot spots has recently been studied using scanning near-field optical spectroscopy, which directly shows that these hot spots are highly localized on a 10-nm scale[23]. The narrow spectral line widths suggest long dephasing times $T_2$ of ~20 fs of hot spots[23]. Their high quality factors (up to 40) and large field enhancements make them appealing for plasmon–emitter coupling. Particularly, different types of quantum emitters can be readily infiltrated into the nanopores and thus be placed in the hot spots, without the need for lithography methods. Furthermore, the broad spectral variation of the hot spot resonances offers an opportunity to probe the effect of detuning on the coupling between plasmons and emitters.

To time resolve the surface plasmon fields of the nanosponge, we performed interferometric frequency-resolved autocorrelation (IFRAC) measurements (Fig. 1a)[24–26]. The sample is excited by a pair of broadband, phase-locked 8-fs pulses, time-delayed by $\tau$. The pulses are centered at ~880 nm and have a nearly transform-limited time structure $E_L(t)$ (Supplementary Fig. 4). The resulting local plasmon field $E_P(t)$ generates a second-order nonlinear field $E_{NL}(t, \tau) \propto (E_P(t) + E_P(t - \tau))^2$. The signal at detection angular frequency $\omega_d$ is resolved in a spectrometer and recorded as a function of $\tau$, giving an IFRAC trace, $I_{IF}(\omega_d, \tau) = |\int E_{NL}(t, \tau)e^{i\omega_d t}dt|^2$. Figure 1b depicts such a trace from a single gold nanosponge, showing broadband nonlinear emission centered at $\omega_d \sim 4.4$ fs$^{-1}$ (~430 nm) that is modulated periodically along the delay axis $\tau$. We find a strong and long-lived (>25 fs) coherent second-harmonic (SH) signal, superimposed on a weak incoherent two-photon photoluminescence background (Supplementary Figs. 5 and 7)[25].

It has previously been shown[24] that the time structure of the field that is driving the nonlinear emission can be retrieved from a Fourier transform of the IFRAC signal along $\tau$, giving a new delay frequency $\omega_\tau$. This Fourier transform shows five distinct bands, a zeroth-order (DC) band around $\omega_\tau = 0$, two fundamental (FM) bands around $\omega_\tau = \pm\omega_d/2$ and two second-order bands around $\omega_\tau = \pm\omega_d$ (Supplementary Fig. 8)[24,25]. Both DC and FM bands contain the desired information about the time profile of $E_P(t)$ (ref. [24]). Figure 1c shows the DC band of the IFRAC trace in Fig. 1b. It appears to be clearly asymmetric and stretched along the $\omega_d$ axis. This is surprising since the SH emission from a single mode would result in a symmetric, Lorentzian line shape of the DC band with the same, homogeneously broadened linewidth of $4/T_2$ along both $\omega_d$ and $\omega_\tau$ axes (Supplementary Figs. 8 and 9). Even in the case of an inhomogeneous ensemble of coherently interfering nonlinear emitters, the DC band should also be symmetric, with a width given by the inhomogeneous linewidth (Supplementary Fig. 10)[27]. This is in contrast to the distinct asymmetry of the DC band seen in Fig. 1c. To explain this observation, we consider that the SH polarization from plasmonic hot spots in disordered metallic nanostructures is usually generated from an Angstrom thin sheet of nonlinear dipole emitters, orientated perpendicularly to the complex-shaped and highly curved metal surface[28,29]. In such a case, one expects that SH emissions from neighboring hot spots do not interfere and that the total SH intensity is given as the incoherent sum of the SH intensities of each hot spot, $I'_{IF}(\omega_d, \tau) = \sum_j I_{IF,j}(\omega_d, \tau)$, with $j$ being the hot spot index[29,30].

When considering this in the modeling of the IFRAC traces, we can reasonably well account for the asymmetric DC band (Fig. 1d), assuming an inhomogeneously broadened distribution of several hot spot modes (nonlinear emitters) with comparatively long $T_2$ times of ~16 fs (see simulated IFRAC and more details in

Supplementary Fig. 13). The $T_2$ time is obtained by fitting the DC band signal at a given $\omega_d$ to a homogeneously broadened line shape model (Fig. 1c, details in Supplementary Fig. 11). The deduced $T_2$ times are close to those deduced using near-field linear scattering spectroscopy of individual hot spots[23].

To confirm this observation, we recorded SH spectra at a fixed $\tau \sim 75$ fs for different polarization angles of the linearly polarized incident light (Fig. 1e, completed data in Supplementary Fig. 5). The SH emission shows a rich dependence on polarization angle, with very broad and multi-peaked spectra. The polar plot of the SH intensity is indicative of a multipolar emission character (Fig. 1f), as expected for the emission from multiple randomly oriented localized plasmonic hot spots at the surface of a single nanosponge. The dipolar mode of the nanosponges can couple efficiently to the far-field light because of their submicron size that matches to the incident wavelength. The dipolar mode then couples to and excites localized hot spot modes. The linear and nonlinear fields from the hot spots can then couple back to the dipolar mode and radiate into the far field. Since the hot spot fields are much stronger than that of the dipole mode, the nonlinear emission is mainly governed by the hot spots.

**Photoelectron emission from individual plasmonic hot spots.** The IFRAC measurements suggest that the nonlinear emission stems from multiple localized hot spots on the surface of the nanosponges, but are unable to resolve emission from a single hot spot. For this, we employed interferometric time-resolved photoemission electron microscopy (tr-PEEM)[31–36]. Similar to IFRAC, the sample is excited by a pair of few-cycle pulses, centered at ~800 nm. Photoemitted electrons are detected, providing sub-100 nm spatial resolution to isolate individual hot spots. Figure 2a shows a SEM image of the gold nanosponge used in this experiment. The tr-PEEM images measured at different $\tau$ are depicted in Fig. 2b, from which three individual hot spots are well resolved. Supplementary Videos 1 and 2 show $\tau$-dependent photoelectron emission from the nanosponge in Fig. 2a and other nanosponges. We note that the PEEM image at $\tau = 0$ does not allow us to clearly isolate the individual hot spots, because of the overlapping emission from the delocalized dipole mode. Electron emission from this short-lived dipole mode can only be observed for short time delays, whereas for long time delays only the localized hot spot modes survive. Interferometric autocorrelation (IAC) traces of the number of detected electrons recorded at each of those hot spots are shown in Fig. 2c, displaying long-lived coherent oscillations, persisting well beyond the time resolution of the experiment of 5–6 fs (ref. [32]). The pronounced fringe contrast is consistent with a high order ($n = 2.5−3$) nonlinear photoemission process. The distinctly different oscillation periods are a clear signature of photoemission from single hot spots with different resonance frequencies[33]. The IAC traces are reasonably well modeled by assuming nonlinear photoemission from a homogenously broadened hot spot emitter with a $T_2$ time of ~13 fs, superimposed by the weak and short-lived field associated with the excitation of the delocalized dipole mode of the entire nanosponge[22]. The fitted $T_2$ times of hot spots are close to those deduced from IFRAC. The tr-PEEM results thus directly confirm that each hot spot acts as an independent emitter that can be selectively excited by tuning the inter-pulse delay.

**Enhanced excitonic emission from Au/ZnO hybrid nanosponges.** We now make use of this selectivity to study the coupling of individual hot spots to nonlinear quantum emitters, by depositing a thin layer of ZnO onto the nanosponge surface[37]. The material is chosen because of its large second-order susceptibility and sharp exciton absorption resonance that is spectrally

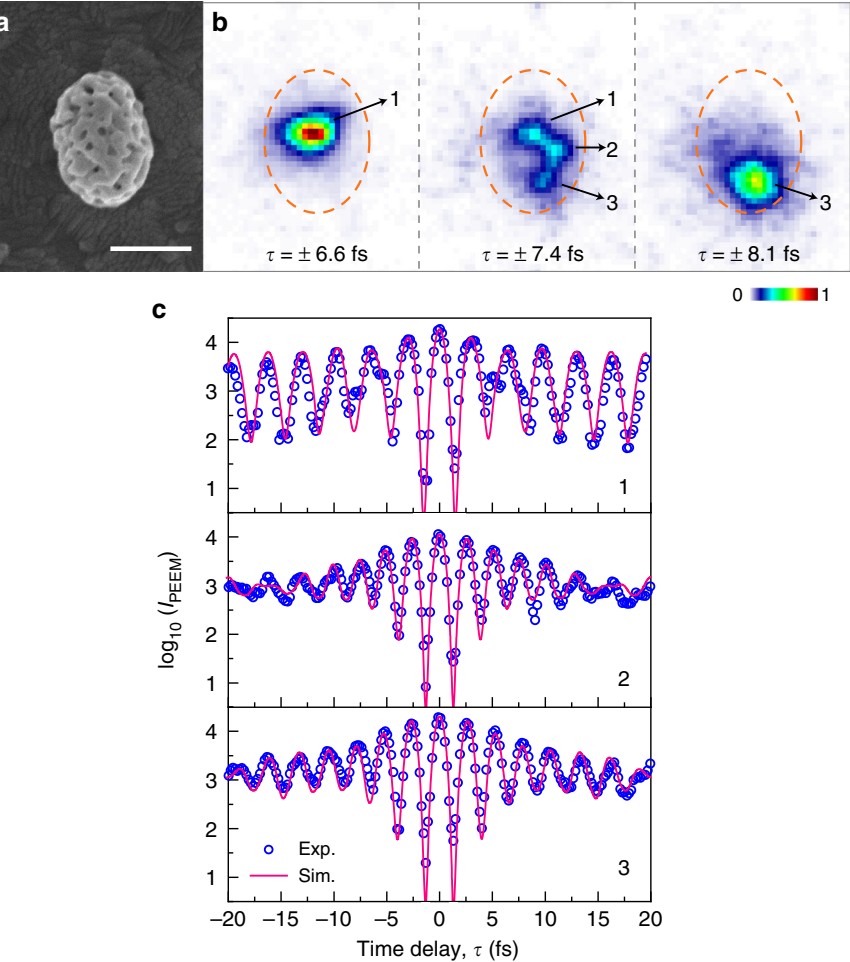

**Fig. 2 Interferometric time-resolved photoelectron emission from individual hot spots of a single gold nanosponge. a** Scanning electron microscopy image of a single gold nanosponge, scale bar: 300 nm. **b** Photoemission electron microscopy images of the nanosponge with its contour indicated by dashed ellipses, recorded at different time delay $\tau$ of the few-cycle pulse pair. Electron emission from three well-defined localized plasmonic hot spots is observed, which can be selectively excited by tuning $\tau$. **c** Experimental (blue circles) and simulated (pink lines) interferometric autocorrelation traces of the electron emission intensity at the three hot spots. Each localized hot spot is modeled as a Lorentzian oscillator that is excited by the laser pulses and a $n$th-order ($n = 2.5 - 3$) nonlinear photoemission process is assumed. For each hot spot, the local field is given as the superposition between the hot spot field and a short-lived (dephasing time $T_{2,dp} = 3$ fs) dipole mode field, delocalized over the nanosponge[22]. The fitting gives the following parameters: resonance frequencies $\omega_{P,j} = 2.02$, 2.48, and 2.37 fs$^{-1}$, dephasing times $T_{2,j} = 17$, 10, and 13 fs, order of nonlinearity $n_j = 3$, 2.5, and 2.7, for hot spots 1–3, respectively.

overlapping with the broad SH emission from the ensemble of plasmonic hot spots. SEM and transmission electron microscopy (TEM) images of such a hybrid Au/ZnO nanosponge (Fig. 3a, more in Supplementary Fig. 2) show that the ZnO material nicely fills the interior of the nanopores and covers the entire gold ligament surface with a 10-nm thick layer.

An IFRAC trace of a single hybrid nanosponge is shown in Fig. 3b. A broad emission is observed around $\omega_{d,P} \sim 4.3$ fs$^{-1}$ (~440 nm), very similar to that seen for the bare gold nanosponge in Fig. 1b. We thus assign this band mostly to plasmonic SH emission. Additionally, a distinctly enhanced emission at the exciton frequency of ZnO, $\omega_X \sim 4.8$ fs$^{-1}$ (~390 nm)[25,37,38], is observed. This band therefore reflects excitonic emission from ZnO and is almost one order of magnitude stronger than the emission from the bare gold nanosponge at the same frequency (Supplementary Figs. 6 and 16). Nonlinear optical spectra taken from many individual ZnO-infiltrated nanosponges confirm this enhancement effect[37]. Spectrally integrated IAC traces $\bar{I}_{IF}(\tau) = \int I_{IF}(\omega_d, \tau) d\omega_d$ from the plasmonic band between 4.1 and 4.65 fs$^{-1}$ show coherent oscillations persisting for >25 fs (Fig. 3c,

bottom), indicating SH emission from long-lived plasmonic hot spots. Interestingly, the IAC trace from the excitonic peak (4.65–4.95 fs$^{-1}$) displays a characteristic beating pattern, reflecting a coherent superposition of at least two modes (Fig. 3c, top). Notably, the IAC trace of the excitonic emission is slightly longer-lived (>30 fs) than the nonlinear plasmonic emission, reflecting the coupling to the sharp ZnO exciton resonance, as will be further discussed later. The DC band of the IFRAC trace has a similarly stretched shape as that of the bare gold nanosponge, again a feature of nonlinear emission from multiple hot spots (Supplementary Fig. 12).

**Quantum pathways of nonlinear plasmon–exciton coupling.** The FM band of the IFRAC trace in Fig. 3b is shown in Fig. 4a, plotting the nonlinear emission intensity as a function of detection ($\omega_d$) and excitation frequency ($\omega_\tau$), which is obtained by Fourier transform along $\tau$. This two-dimensional (2D) FM spectrum allows us to visualize the effect of a selective excitation of different hot spots (vertical arrows) on the nonlinear emission (horizontal arrows). In

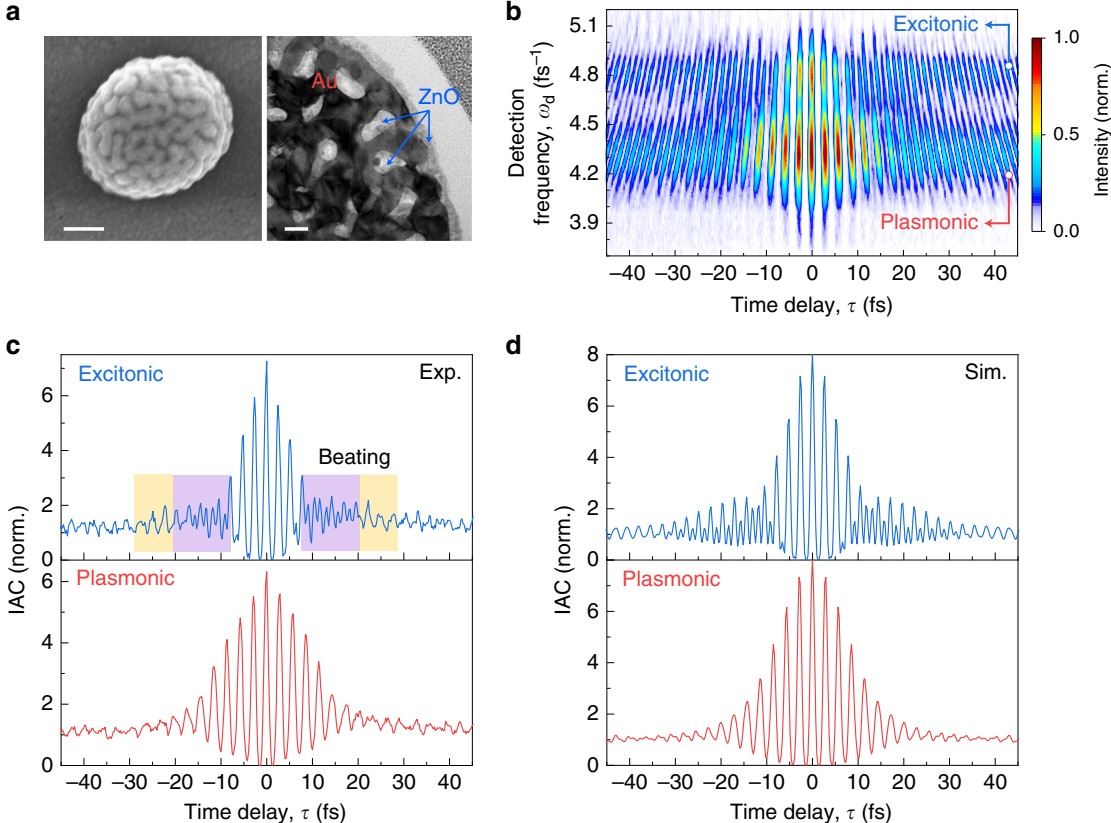

**Fig. 3 Enhanced coherent excitonic emission from a single hybrid Au/ZnO nanosponge. a** Scanning electron microscopy (left, scale bar: 100 nm) and cross-sectional transmission electron microscopy (right, scale bar: 20 nm) images of a hybrid Au/ZnO nanosponge. A 10-nm thick layer of ZnO has been deposited onto their outer surface and infiltrated into the nanopores. **b** IFRAC trace of a single hybrid Au/ZnO nanosponge. Two distinct emission bands can be observed: a broad nonlinear plasmonic emission at ~4.3 fs$^{-1}$ and a distinctly enhanced excitonic emission at ZnO exciton frequency of ~4.8 fs$^{-1}$. **c** Experimental interferometric autocorrelation (IAC) traces integrated over the range of the plasmonic (bottom) and excitonic (top) emission from the IFRAC trace in **b**. The beating pattern showing destructive (purple region) and constructive (yellow region) interference at the excitonic emission band results from SF generation from a single hot spot. **d** IAC traces simulated by a nonlinear plasmon–exciton coupling model, agreeing well with the experiment.

the plasmonic emission range, the signal is distributed along the diagonal line, $\omega_d = 2\omega_\tau$, with a slope of 2, indicating a SH process. Here, the selective excitation of hot spots with resonant frequencies $\omega_{P,j}$ ($j$ is the hot spot index) leads to SH emission at $2\omega_{P,j}$.

In the excitonic emission range, we observe a comparatively large intensity, about four times stronger than that in the 2D-FM spectrum of the bare gold nanosponge (Supplementary Figs. 7 and 16). Surprisingly, the signal splits into two side peaks, symmetrically positioned around the diagonal. Analytical analysis suggests that this split is the characteristic of SF generation, $\omega_d = \omega_1 + \omega_2$, i.e., the nonlinear mixing of different linear fields at $\omega_1$ and $\omega_2$ (see a completed analytical formalization in Supplementary Note 7). The horizontal cross-sectional spectrum at $\omega_X \sim 4.8$ fs$^{-1}$ shows that the excitonic emission is mainly SF generated from fundamental modes at $\omega_1 \sim 2.3$ fs$^{-1}$ and $\omega_2 \sim 2.5$ fs$^{-1}$, whereas SH generation from 2.4 fs$^{-1}$ mode is weaker. This splitting also explains the beating of the IAC trace at $\omega_X$ in Fig. 3c, reflecting the coherent superposition of those two distinct fundamental modes. The 2D-FM spectrum thus nicely correlates fundamental excitation to second-order emission frequencies (see more detailed discussion and prove in Supplementary Note 7). The resonances and their line shapes in this map provide an opportunity to reveal the quantum pathways of the enhanced excitonic emission, which can either be excited through direct light–exciton coupling or driven by plasmonic near fields via

linear or nonlinear plasmon–exciton coupling. We shall examine each of the three quantum pathways.

First, we assume that ZnO excitons are excited only by far-field light. In case of a narrowband laser excitation, efficient excitonic emission can only be achieved via SH generation when tuning the laser to half of the exciton frequency. In the 2D-FM spectrum, this pathway will show up as a peak at the diagonal, centered at ($\omega_X/2$, $\omega_X$), in stark contrast to the experiment. For a broadband, few-cycle laser excitation, excitonic emission can also be triggered by many SF channels, mixing a pair of colors $\omega_1$ and $\omega_2$ from the laser spectrum such that $\omega_1 + \omega_2 = \omega_X$. Hence, for each pair, we will see two split peaks at ($\omega_1$, $\omega_X$) and ($\omega_2$, $\omega_X$) in the 2D-FM spectrum. A simulation of such a 2D spectrum using a classical nonlinear oscillator model and the time profile of our laser is shown in Supplementary Fig. 17. We find that the cross-sectional spectrum $I(\omega_\tau)$ at $\omega_d = \omega_X$ extends over the entire bandwidth of our laser and has a spectral width that is much wider than the experimental data. Far-field laser excitation therefore cannot account for the experiments, in agreement with observing no detectable signal from a pure, 10-nm thick ZnO layer under the same experimental conditions.

Secondly, we consider plasmon–exciton coupling. Here, the far-field laser first couples to the delocalized, dipolar plasmon mode of the nanosponge, which resonantly excites different plasmonic hot spots at the nanosponge surface[22,23]. This strong hot spot field,

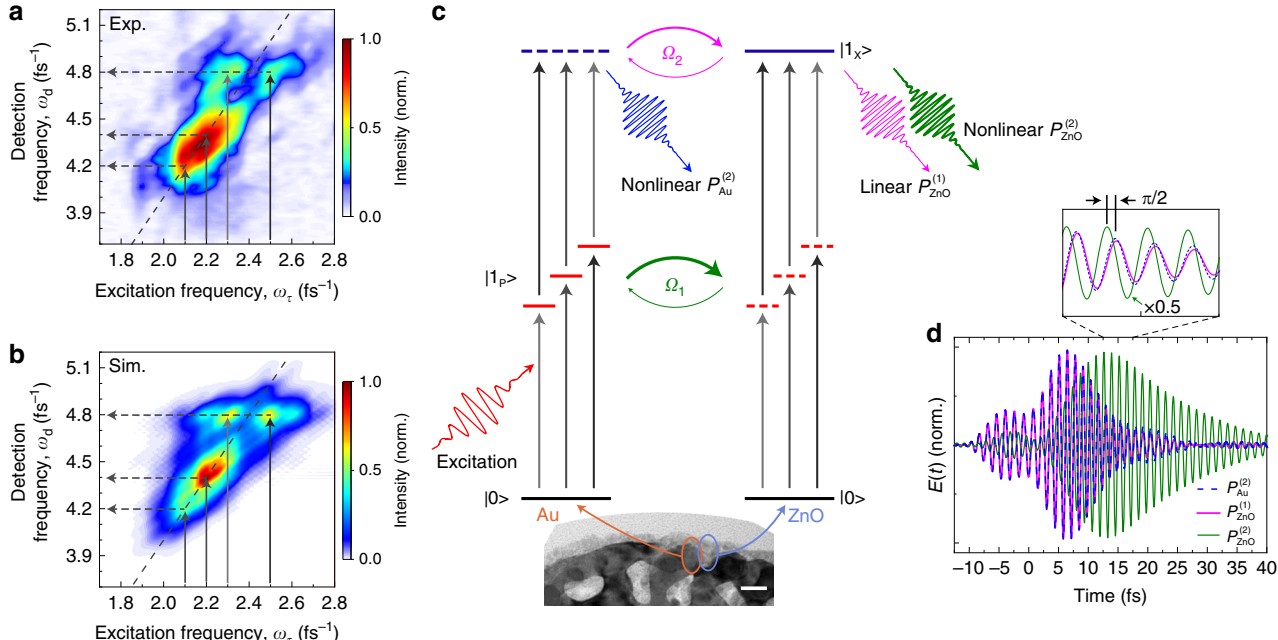

**Fig. 4 Two-dimensional spectra reveal quantum pathways of nonlinear plasmon–exciton coupling. a** Fourier transformed fundamental band of the IFRAC trace in Fig. 3b, correlating fundamental excitation (vertical arrows) to nonlinear emission signals (horizontal arrows). The signal is symmetric with respect to the diagonal line with a slope of 2, indicating a second-order nonlinear process. The split signal at the exciton frequency shows that the excitonic emission is SF generated ($\omega_X = \omega_1 + \omega_2$) with two fundamental modes at $\omega_1 \sim 2.3$ fs$^{-1}$ and $\omega_2 \sim 2.5$ fs$^{-1}$. The nonlinear plasmonic emission signal at the diagonal line corresponds to second-harmonic generation ($\omega_d = 2\omega_{P_j}$). **b** Simulated fundamental band using a nonlinear plasmon–exciton coupling model, reproducing well the features of the experiment. **c** Schematic illustration of nonlinear plasmon–exciton coupling. Few-cycle pulses excite several plasmonic hot spot modes $|1_P\rangle$ with distinct resonance frequencies at the surface of the hybrid nanosponge (bottom inset: transmission electron microscopy image, scale bar: 20 nm), generating a nonlinear plasmonic polarization $P_{Au}^{(2)}$. In second-order coupling ($\Omega_2$), $P_{Au}^{(2)}$ are coherently Rayleigh scattered by the ZnO inclusion, giving a linear polarization $P_{ZnO}^{(1)}$. In first-order coupling ($\Omega_1$), the locally enhanced plasmonic fields drive the second-order nonlinear polarization $P_{ZnO}^{(2)}$ of the ZnO exciton $|1_X\rangle$. In all quantum pathways, the nonlinear fields can be generated via either second-harmonic or SF channels (marked by gray arrows with their lengths representing different energies). **d** Simulated electric fields of the three emission pathways for a coupled hot spot–exciton system from a single plasmonic hot spot at $\omega_P = 2.5$ fs$^{-1}$. After scaling, $P_{ZnO}^{(1)}$ is almost identical and in-phase with $P_{Au}^{(2)}$. In contrast, $P_{ZnO}^{(2)}$ is phase-shifted by about $-\pi/2$ and prolonged in time with respect to $P_{Au}^{(2)}$.

$E_P^{(1)}$ generates a second-order nonlinear polarization at the gold surface, $P_{Au}^{(2)}$, accounting for the nonlinear emission in the bare gold nanosponge (Fig. 4c, left). In the hybrid nanosponge, this nonlinear plasmonic field can be coherently scattered into the far field by the ZnO inclusion, inducing Rayleigh scattering, i.e., a linear dipole polarization $P_{ZnO}^{(1)}$ (via second-order coupling, $\Omega_2$, magenta arrows in Fig. 4c). This enhances the amount of light that is scattered into the far field. An estimate based on a simplified coupled dipole model (see details in Methods section and Supplementary Note 8) suggests a maximum intensity enhancement factor of $\left|\frac{3\varepsilon(\omega)}{\varepsilon(\omega)+2}\right|^2$. The dielectric function of ZnO, $\varepsilon(\omega) = \varepsilon_B + \varepsilon_X(\omega)$, is dominated by a strong, almost frequency-independent background $\varepsilon_B \approx 4-5$ and a much weaker resonant exciton contribution with a magnitude $|\varepsilon_X| \approx 1$ (ref. [39]). Due to this strong background, the on-resonant enhancement factor of the scattering intensity at $\omega_X$ should be only ~10% stronger than the off-resonant enhancement factor at $\omega_{d,P}$ (Supplementary Note 8). That is, the enhancement is almost frequency independent. This contrasts with the experimentally observed ratio of on- and off-resonant enhancement factor of >350% (Supplementary Fig. 16). Therefore, the scattering of the nonlinear plasmonic field by a ZnO inclusion alone cannot explain the large resonant enhancement at $\omega_X$. Also, a change of the linear optical properties of the hot spot modes, specifically their linewidth and frequency[11,13], due to the ZnO inclusion, does not account for this

resonant enhancement since it would affect the on- and off-resonant nonlinear signal in exactly the same way[13].

We are led to the third pathway: the linear plasmonic field instead couples to ZnO excitons via first-order coupling ($\Omega_1$, green arrows in Fig. 4c) by driving the exciton nonlinearly. Here, the part of $E_P^{(1)}$ that is spatially overlapping with the ZnO inclusion can off-resonantly drive a second-order nonlinear polarization from ZnO, $P_{ZnO}^{(2)}$. Earlier measurements have shown that the SH emission from ZnO thin films is strongly enhanced near the ZnO bandgap with a second-order susceptibility that is increased by more than a factor of five when approaching the exciton resonance[40]. Therefore, the nonlinear signal $P_{ZnO}^{(2)}$ will mostly be centered around $\omega_X$ (refs. [25,38]), showing excitonic enhancement that is consistent with the experiment. We therefore consider this first-order coupling as the main mechanism for the enhanced excitonic emission. The time structure of $P_{ZnO}^{(2)}$ will be given by a convolution of $E_P^{(1)}$ with the resonant excitonic response of ZnO. In comparison to $P_{Au}^{(2)}$, $P_{ZnO}^{(2)}$ thus displays a characteristic phase shift by ~$\pi/2$ arising from the resonant response of the exciton to the plasmonic driving field (see detailed phase analysis in Supplementary Note 9), and is considerably prolonged in time due to the persistent coherence of the excitonic transition. This is seen in the simulated electric fields for the excitation of a single hot spot at $\omega_P = 2.5$ fs$^{-1}$ in Fig. 4d (blue and green curves). In contrast, the linearly scattered field $P_{ZnO}^{(1)}$ (pink

curve in Fig. 4d) is in-phase with and shows almost the same dynamics as $P_{Au}^{(2)}$, resulting from the dominant dielectric background contribution as discussed above. The optical dynamics of the plasmon-driven nonlinear excitonic emission therefore depends on both the optical dynamics of the fundamental driving hot spots and the finite dephasing time of the exciton transition, as reflected by the elongated IAC trace of the excitonic emission compared to that of the plasmonic emission in Fig. 3c.

With this, we can now successfully understand the experimental results. To describe the nonlinear emission from the entire hybrid nanosponge, we assume several well-separated hot spots at its surface with different resonance frequencies ($\omega_{P,j} = 2.0 - 2.6 \, \text{fs}^{-1}$) and dephasing times of ~16 fs. Direct evidence for the existence of these hot spot resonances, their long dephasing times and small spatial localization lengths of ~10 nm is given by scanning near-field optical spectroscopy[23] and PEEM (Fig. 2) measurements. Each of these hot spots is driven by the laser field and coupled to neighboring ZnO excitons. We consider all three contributions to the nonlinear emission depicted in Fig. 4c (details of the model in Methods section). The simulated IFRAC trace (Supplementary Fig. 19), 2D-FM spectrum (Fig. 4b), and IAC traces (Fig. 3d) can reasonably well account for the experiments, as long as we assume that plasmon-driven nonlinear excitonic emission, $P_{ZnO}^{(2)}$ dominates the emission around $\omega_X$. Indeed, as in the experiment, the model predicts a splitting of the 2D-FM spectrum near $\omega_X$ into two excitation peaks around $\omega_1 = 2.3 \, \text{fs}^{-1}$ and $\omega_2 = 2.5 \, \text{fs}^{-1}$, suggesting that SF generation dominates the nonlinear response in this frequency range.

## Discussion

In principle, for a single plasmonic eigenmode, such a splitting is unlikely to occur for a smooth laser spectrum, which would favor SH generation from hot spot fields resonant at $\omega_X/2$. This SH channel, however, is not efficiently excited by our laser, which shows a slight dip in the spectrum at $\omega_X/2 \sim 2.4 \, \text{fs}^{-1}$ (Supplementary Figs. 4 and 20), thus suppressing the coupling of this hot spot field to the ZnO exciton. The spectral structure of our laser therefore results in nonlinear excitonic emission that is off-resonantly driven by two different frequency components $\omega_1$ and $\omega_2$ of a hot spot field such that $\omega_1 + \omega_2 = \omega_X$. As seen in the 2D-FM spectra, this new, SF channel is efficient if the hot spot resonance is reasonably close to $\omega_X/2$. This SF channel is particularly helpful to enhance the overall yield of nonlinear emission in realistic samples with a large number of hybrid plasmon–exciton antennas, where disorder-induced fluctuations vary the resonance conditions from one antenna to another. In such systems, efficient nonlinear emission can be achieved via SF generation by excitation with ultrashort broadband laser pulses, even if the plasmon resonance is detuned from $\omega_X/2$, for which SH generation with narrowband excitation is inefficient.

We note that for more detuned hot spots the coupling to the ZnO excitons is reduced, as seen in Fig. 4a. This arises because the two fields for SF mixing are most likely created by the same hot spot eigenmode, and thus only one of the two fields, $\omega_1 \approx \omega_P$, will be resonantly enhanced (Supplementary Fig. 20). To maximize the efficiency of SF generation, one could use multifrequency antennas with two or more energetically detuned linear plasmonic eigenmodes that are spatially partially overlapping with the same hot spot. Here, more frequency components for SF mixing can be resonantly excited and can coherently interact with the same nonlinear emitter placed in the hot spot. Typical sample geometries of multifrequency plasmonic resonators are two-arm antennas with different arm lengths[41], cross antennas with two pairs of nanorods perpendicular to each other[11,42], and other tailored sample geometries[6,43]. These structures support spatially overlapping and thus coherently interacting plasmonic modes that can lead to efficient nonlinear SF coupling to excitons. SF generation thus allows for highly tunable nonlinear plasmon–exciton coupling in a variety of hybrid nanostructures. Our result suggests new SF quantum channels for nonlinear plasmon–exciton coupling. Compared to commonly adopted SH quantum channels, SF generation offers much wider spectral tuneability for the coherent manipulation of the linear and nonlinear fields, opening up new routes for optical control in such a nonlinearly coupled nanosystem. The results also opens up the possibility to explore difference frequency generation $|\omega_1 - \omega_2|$ of two plasmonic fields for coupling to low-frequency infrared or terahertz vibrations, for example, for plasmon-enhanced stimulated Raman scattering[44].

In summary, we have spatially and temporally resolved individual plasmonic hot spot modes at the surface of a gold nanosponge, and studied their nonlinear coupling to ZnO excitons by probing the ultrafast optical dynamics in a hybrid Au/ZnO nanosponge. We demonstrate how fundamental plasmonic fields couple to higher-energy excitons via SF mixing, enhancing coherent excitonic emission. The much prolonged excitonic fields may provide a valuable tool for coherent control via nonlinear plasmon–emitter coupling[20]. Our results not only allow us to unravel the different microscopic contributions to the nonlinear signal enhancement, but may also prove helpful for the design of a new class of hybrid nanostructures with coupled, phased arrays of plasmonic antennas[15], and quantum emitters with strong optical nonlinearities.

## Methods

**Preparation of nanosponges**. Gold nanosponges were synthesized in two steps[21,45]. First, gold–silver alloy nanoparticles with (sub)micron diameters were fabricated by solid-state dewetting of a bimetallic Au/Ag (8 nm/20 nm thick) thin film on a SiO$_2$/Si substrate. Subsequently, the nanoparticles were dealloyed in which silver was chemically removed in a solution of HNO$_3$ (65 wt%), leading to nanoporous structure with 10-nm sized channels perforating the entire particle[22]. To prepare the hybrid nanosponge, a ZnO layer of 10-nm thickness was deposited into the nanopores of bare nanosponges using plasma-enhanced atomic layer deposition at 150 °C. The samples were annealed at 500 °C in Ar for 1 h to improve the crystallinity of ZnO. The nanosponges were transferred to indium tin oxide substrate for measurements.

**IFRAC microscopy**. We used a home-built nonlinear microscopy setup[38] combined with broadband, 8-fs pulses from a 80-MHz Ti:Sapphire oscillator (Venteon: ultra) centered at ~880 nm. A phase-locked pulse pair is prepared in a dispersion-balanced Mach–Zehnder interferometer with inter-pulse time delay $\tau$ controlled by a piezo scanner (Physik Instrumente P-621.1CD PIHera). The laser is focused onto the sample to a diffraction-limited spot size of 1.0 μm through an all-reflective Cassegrain objective with a numerical aperture of 0.5, preserving the time structure of the pulses[46]. The laser power at the focus is ~0.45 mW at $\tau = 0$. The pulse in the focus has been characterized by IFRAC measurements on a 15-μm thick beta barium borate crystal and a retrieval algorithm was used to retrieve the electric field of the pulse (Supplementary Note 2), showing a nearly transform-limited time structure. A half-wave plate is used to tune the polarization of the incident pulse. The back-scattered light is spectrally filtered by a dichroic mirror and then short-pass filtered to remove the linear scattering signal. The nonlinear signal is spectrally dispersed in a monochromator (Princeton Instruments Acton SpectraPro-2500i) and detected with a liquid nitrogen cooled CCD camera (Princeton Instruments Spec-10).

**PEEM measurements and fitting**. Broadband 6-fs laser pulses centered around 800 nm are amplified in an optical parametric chirped pulse amplification laser system at a repetition rate of 200 kHz. They are split into a pulse pair in a dispersion-balanced Mach–Zehnder interferometer and focused onto the gold nanosponge sample in an ultra-high vacuum chamber, using a lens with a focal length of 20 cm under an angle of 65° to the surface normal (p-polarized)[32]. The laser pulse energy is attenuated such that only single electrons are emitted from the sample per laser shot. The emitted electrons are collected with the extractor lens of a PEEM (Focus GmbH) and imaged with a spatial resolution of ~50 nm. The time profile of the laser pulses $E_L(t)$ is characterized by the dispersion scan and frequency-resolved optical gating techniques close to the vacuum chamber with a replica of the chamber window and the focusing lens in the beam path. The extracted spectral phase from both techniques is very similar and confirms a nearly

transform-limited pulse duration of 6.0 fs (full-width at half-maximum of the intensity profile).

To model the IAC traces of the electron emission shown in Fig. 2c, the local electric field at the position of each hot spot $E_{S,j}(t)$ is given as the convolution of the laser pulse and the local response function, which is taken as the sum of a long-lived hot spot mode (subscript $j$ being the hot spot index) and a short-lived dipolar mode (subscript dp)[22],

$$E_{S,j}(t) = E_L(t) \otimes \left\{ \Theta(t) \left( a_j\, e^{-i\omega_{P,j}t} e^{-\frac{t}{T_{2,j}}} + a_{dp}\, e^{-i\omega_{dp}t} e^{-\frac{t}{T_{2,dp}}} \right) \right\} \quad (1)$$

where $\omega$ and $T_2$ are the central frequency and dephasing time, respectively, of the plasmonic modes (hot spots or dipole mode). $\Theta(t)$ is the Heaviside function. The amplitudes of the hot spot mode and dipole mode are represented as $a_j$ and $a_{dp}$, respectively. The IAC trace of each hot spot is then modeled as

$$C_j(\tau) = \int \left| \left( E_{S,j}(t) + E_{S,j}(t-\tau) \right)^{n_j} \right|^2 dt, \quad (2)$$

where $n_j$ is the order of nonlinearity for photoemission.

In the experiment, we can clearly observe a change of the central wavelength when varying the time delay. At early time delays, both delocalized dipole mode and localized hot spot modes contribute, whereas at later time delays only the hot spot modes survive. Since the PEEM signal depends in a highly nonlinear manner on the local electric field amplitude ($I_{PEEM} \propto E^6$), both fields must be of very similar amplitude ($a_{dp} \approx a_j$) to make both contributions significant to the signals at early time delays. For fitting the experimental IAC trace, the amplitude of the dipole mode $a_{dp}$ is kept the same for each simulation of $E_{S,j}$ of the three hot spots in Fig. 2. We find that the amplitudes $a_j$ of the hot spots 1–3 are 1.1, 0.84, and 0.83 as large as $a_{dp}$, respectively.

The fitting gives the following parameters: $\omega_{P,j} = 2.02$, 2.48, 2.37 fs$^{-1}$, $T_{2,j} = 17$, 10, 13 fs, and $n_j = 3$, 2.5, 2.7 for hot spots 1–3, respectively. For the dipole mode, $\omega_{dp} = 2.35$ fs$^{-1}$ and $T_{2,dp} = 3$ fs. Supplementary Figure 21 shows the response functions of the dipole mode and the three hot spot modes, and their convoluted fields with laser in time and frequency domain. We note that the observed order of nonlinearly is slightly lower than estimated from a photon picture needed for the multiphoton-emission of electrons from gold. This may be explained by (i) the applied DC bias voltage bending the vacuum level and lowering the work function[47], (ii) emission from local surface (adsorbate) states closer to vacuum level, and (iii) emission from a hot, nonequilibrium electron system with an effective electron temperature well above room temperature[47].

**Model of nonlinear plasmon–exciton coupling.** We use a classical nonlinear oscillator model to simulate the second-order nonlinear emission from the sample by modeling plasmonic hot spots and exciton resonances as Lorentzian oscillators[48]. We consider three distinct quantum pathways that contribute to the nonlinear emission from hybrid Au/ZnO nanosponges.

The first pathway is the nonlinear plasmonic emission from each of the plasmonic hot spots, which can be excited by an external laser field $E_L(t)$,

$$\ddot{x} + 2\gamma_P \dot{x} + \omega_P^2 x + a_P x^2 = -eE_L(t)/m, \quad (3)$$

where the displacement of the induced collective charge oscillations is denoted as $x$, and $-e$ and $m$ are electron charge and mass, respectively. The resonance frequency of the plasmonic hot spot is $\omega_P$ and its damping rate $\gamma_P = 1/T_{2,P}$, with $T_{2,P}$ being the dephasing time of the hot spot. The strength of the second-order nonlinearity of the hot spot is given as $a_P$ which relates to the second-order nonlinear susceptibility $\chi_P^{(2)} \propto a_P$. We solve Eq. (3) in two steps using perturbation theory[48],

$$\ddot{x}^{(1)} + 2\gamma_P \dot{x}^{(1)} + \omega_P^2 x^{(1)} = -eE_L(t)/m, \quad (4)$$

$$\ddot{x}^{(2)} + 2\gamma_P \dot{x}^{(2)} + \omega_P^2 x^{(2)} + a_P (x^{(1)})^2 = 0. \quad (5)$$

Here, superscripts (1) and (2) represent the linear and nonlinear electronic motion, respectively. Equation (4) gives the linear polarization $P_{Au}^{(1)}(t) \propto x^{(1)}(t)$ and Eq. (5) generates the second-order surface polarization $P_{Au}^{(2)}(t) \propto x^{(2)}(t)$. For gold plasmons, the first-order perturbation is resonant at the frequency of the fundamental plasmon mode ($\omega_P \sim 2.3$ fs$^{-1}$) and the second-order perturbation is off-resonant. This is different from the infiltrated ZnO, for which the polarization is resonantly enhanced at frequencies around the ZnO bandgap (~5.0 fs$^{-1}$). Therefore, fields emitted from ZnO in the frequency range of the fundamental plasmon mode are comparatively weak and can safely be neglected. Also, the back-coupling of the resonant emission around the ZnO bandgap to the plasmonic hot spots is neglected since it is largely detuned from the plasmon resonance.

The second pathway is the nonlinear excitonic emission from ZnO. ZnO is a direct bandgap semiconductor with a bandgap of ~3.3 eV and a quite complex optical response that has been studied in detail during the past decades. For mid-bandgap optical excitation, as in the present experiments, the response is usually dominated by excitonically enhanced SH emission showing a pronounced resonant enhancement near the bandgap[38,40]. Even though more complex responses have been observed for strong optical driving[49], we assume that this second-order nonlinear excitonic emission from ZnO is the dominant contribution to the detected signal in our experiment. It can be excited by (i) the incident far-field laser light $E_L(t)$ or (ii) the local linear and nonlinear optical polarization of plasmonic

near fields $x = x^{(1)} + x^{(2)}$ of neighboring plasmonic hot spots, with coupling strengths $\Omega_1$ and $\Omega_2$, respectively, defining the relation between plasmonic polarization and the field it creates at the position of the ZnO inclusion. The equation of motion follows

$$\ddot{y}^{(1)} + 2\gamma_X \dot{y}^{(1)} + \omega_X^2 y^{(1)} = -e\left( E_L(t) + \Omega_1 x^{(1)} + \Omega_2 x^{(2)} \right)/m, \quad (6)$$

$$\ddot{y}^{(2)} + 2\gamma_X \dot{y}^{(2)} + \omega_X^2 y^{(2)} + a_X (y^{(1)})^2 = 0, \quad (7)$$

where $y^{(1)}$ and $y^{(2)}$ represent linear and second-order nonlinear electronic motion of ZnO, and $\omega_X$ and $\gamma_X$ are resonance frequency, and damping rate of the ZnO exciton, respectively. In contrast to gold plasmons, the linear excitation of ZnO (Eq. (6)) induced by $E_L(t)$ and the linear plasmonic field $\Omega_1 x^{(1)}$ is off-resonant, but is resonantly enhanced near the bandgap by the nonlinear plasmonic field $\Omega_2 x^{(2)}$ via second-order coupling, leading to the Rayleigh scattering as will be discussed in detail in the third pathway later. Importantly, however, the nonlinear excitonic oscillation $y^{(2)}$ is mainly induced by the locally enhanced linear plasmonic field $\Omega_1 x^{(1)}$ via first-order coupling. As discussed in the main text, laser excitation $E_L(t)$ cannot explain the experiment and can be excluded (see simulation result in Supplementary Fig. 17), and the effect of $\Omega_2 x^{(2)}$ is completely off-resonant (note the square in Eq. (7)). The oscillation $y^{(2)}$ induces a nonlinear excitonic polarization $P_{ZnO}^{(2)}(t)$ which emits into the far field, interfering with $P_{ZnO}^{(2)}(t)$ from the same hot spot. As discussed above, the back-coupling of this field to the plasmonic field can be neglected.

The third pathway is the Rayleigh scattering of nonlinear plasmonic field by ZnO. Here, the electromagnetic fields from the ZnO inclusions can also be emitted by resonantly scattering the local nonlinear plasmonic fields $\Omega_2 x^{(2)}$ at the position of the ZnO inclusion. Knowing that the plasmonic fields are emitted from localized hot spots with diameters $2r_P$ of ~10 nm (ref. [23]), we model each hot spot as a small dipole emitter with nonlinear dipole moment $p_{Au}^{(2)} = P_{Au}^{(2)} A_{Au}$, where $A_{Au}$ is the surface area of the hot spot. This dipole moment generates a near field at frequency $\omega$ of $\mathbf{E}(\mathbf{r}, \omega) = \frac{\omega^2}{\varepsilon_0 c^3} \overset{\leftrightarrow}{\mathbf{G}}(\mathbf{r}, \omega) p_{Au}^{(2)}(\omega)$, with $\overset{\leftrightarrow}{\mathbf{G}}$ denoting the near-field term of the Greens' propagator. We model the ZnO inclusion as a small, spherical dielectric nanoparticle with radius $r_X$. In quasi-static approximation, the linear polarizability of the ZnO particle is then given as

$$\alpha(\omega) = 4\pi\varepsilon_0 r_X^3 \frac{\varepsilon(\omega) - 1}{\varepsilon(\omega) + 2} \quad (8)$$

Here, the frequency dependent dielectric function of ZnO, $\varepsilon(\omega) = \varepsilon_B + \varepsilon_X(\omega)$, is dominated by a strong, almost frequency-independent, real-valued dielectric background $\varepsilon_B \approx 4 - 5$, and a much weaker resonant exciton contribution,

$$\varepsilon_X(\omega) = |\varepsilon_X| \frac{\gamma_X}{\omega - \omega_X + i\gamma_X}, \quad (9)$$

with a magnitude $|\varepsilon_X| \approx 1$ (ref. [39]). Taking the free-space Greens' propagator, the maximum local electric field at the center of the ZnO nanoparticle created by the nonlinear plasmonic dipole moment is

$$E(\omega) = \frac{1}{2\pi\varepsilon_0} \frac{p_{Au}^{(2)}(\omega)}{\bar{r}^3} = \Omega_2 x^{(2)}, \quad (10)$$

with $\bar{r}$ being the distance between the center of the nonlinear plasmonic dipole and that of the ZnO nanoparticle. This field is now driving the ZnO oscillator, creating a linear dipole moment

$$p_{ZnO}^{(1)}(\omega) = \alpha(\omega)E(\omega) = \frac{2r_X^3}{\bar{r}^3} \frac{\varepsilon(\omega) - 1}{\varepsilon(\omega) + 2} p_{Au}^{(2)}(\omega). \quad (11)$$

The time profile of the scattered field $p_{ZnO}^{(1)}(t)$ is obtained by Fourier transform $p_{ZnO}^{(1)}(t) = \int p_{ZnO}^{(1)}(\omega)e^{-i\omega t}d\omega$. This contains two contributions, $p_{ZnO}^{(1)}(t) = p_B^{(1)}(t) + p_X^{(1)}(t)$, in which the scattered field $p_B^{(1)}(t)$ by $\varepsilon_B$ is essentially in-phase and has the same time dynamics as that of $p_{Au}^{(2)}(t)$ because $\varepsilon_B$ is a real-valued constant. Instead, the field linearly scattered by exciton polarization $p_X^{(1)}(t) \propto y^{(1)}(t)$ is ~$\pi/2$ phase-shifted with respect to $p_{Au}^{(2)}(t)$ and with elongated time dynamics given by the linear convolution between $p_{Au}^{(2)}(t)$ and the response function of the exciton resonance. The total scattered field $p_{ZnO}^{(1)}(t)$ is, however, dominated by the large $\varepsilon_B$ with much weaker contribution from $\varepsilon_X(\omega)$. This is seen in the simulated field shown as the pink curve in Fig. 4d, which has almost the same profile as $p_{Au}^{(2)}(t)$ (blue curve).

Equation (11) also allows us to estimate the enhancement factor of nonlinear emission near the ZnO bandgap resulting from Rayleigh scattering of $p_{Au}^{(1)}$ by the ZnO inclusion. Optimally, $p_{Au}^{(1)}$ and $p_{ZnO}^{(1)}$ align in parallel, resulting in a total dipole moment $p_{tot}(\omega) = p_{ZnO}^{(1)}(\omega) + p_{Au}^{(1)}(\omega)$ and an enhancement factor of the emission intensity $F_{enh}(\omega) = |p_{tot}(\omega)|^2 / |p_{Au}^{(1)}(\omega)|^2$. To maximize the enhancement, we

choose $\bar{r} = r_X$, i.e., place $p_{Au}^{(2)}$ at the surface of the ZnO nanoparticle, and get

$$F_{enh}(\omega) \approx \left| \frac{3\varepsilon(\omega)}{\varepsilon(\omega)+2} \right|^2. \tag{12}$$

The enhancement factor thus simply depends on the dielectric function of ZnO. Using the refractive index $n(\omega)$ from ref. [39] and $\varepsilon(\omega) = n(\omega)^2$, the on-resonant enhancement factor $F_{enh}^{on}(\omega = \omega_X)$ at exciton frequency is estimated to be ~10% larger than the off-resonant enhancement factor $F_{enh}^{off}(\omega = \omega_{d,P})$ at the main plasmonic emission peak $\omega_{d,P} \sim 4.3\,\mathrm{fs}^{-1}$, see details in Supplementary Note 8.

In this phenomenological model, the total nonlinear emission from a single hot spot, $I_{NL,j}(\omega)$, in the hybrid nanosponges thus stems from the above discussed three distinct pathways. These are added coherently to give $E_{NL,j}(t) = a \cdot x^{(2)}(t) + b \cdot y^{(2)}(t) + c \cdot y^{(1)}(t)$ and $I_{NL,j}(\omega) = \left| \int E_{NL,j}(t) e^{i\omega t} dt \right|^2$. The total emission from an individual hybrid nanosponge is then obtained by incoherently adding up the nonlinear emission from all hot spots, $I'_{NL}(\omega) = \sum_j I_{NL,j}(\omega)$, as discussed in the main text. The amplitude coefficients $a$, $b$, and $c$ are taken as free scaling parameters to achieve acceptable agreement between simulations and experiments.

## Data availability
The data that support the plots within this paper and other findings of this study are available from the corresponding author upon reasonable request.

## Code availability
The MATLAB codes used in this study are available from the corresponding author upon reasonable request.

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

## Acknowledgements

The authors acknowledge financial support by the Deutsche Forschungsgemeinschaft (SPP1839 'Tailored Disorder', grants LI 580/12, RU 1383/5, SCHA 632/24, and SPP1840 'QUTIF' grant LI 580/13), the Korea Foundation for International Cooperation of Science and Technology (Global Research Laboratory project, K20815000003), the German–Israeli Foundation (GIF grant no. 1256 and 1074-49.10/2009), the Swedish Research Council, Laserlab-Europe EU-H2020 654148, and the European Union Horizon 2020 research and innovation program under the Marie Skłodowska-Curie grant agreement 793604 ATTOPIE. J.-H.Z. is supported by Alexander von Humboldt Postdoctoral Fellowship and Carl von Ossietzky Young Researchers' Fellowship from University of Oldenburg. The authors are grateful to Mario Ziegler from IPHT Jena for the help with sample preparation, and Dominik Flock and Dr. Thomas Kups from TU Ilmenau for TEM examination. We thank Moritz Gittinger and Dr. Martin Silies for their help with the measurements of dark-field scattering spectra.

## Author contributions

C.L. initiated and coordinated the study. D.W. and P.S. prepared the samples. J.-H.Z. and J.-M.Y. performed IFRAC measurements with the help of A.K., and A.C. J.-H.Z., J.-M.Y., and C.L. analyzed the data, and performed analytical and numerical simulations of the IFRAC. E.R. helped with theoretical modeling. J.V. initiated the tr-PEEM measurements, set up the experiment, and analyzed the data. J.V. and L.W. performed the tr-PEEM experiments with the help of S.M. and under supervision of C.L.A, A.L.H., and A.M. J.-H.Z. and C.L. co-wrote the manuscript. All authors discussed the results and commented on the manuscript.

## Competing interests

The authors declare no competing interests.

## Additional information

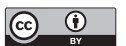

