## [Peer Review File · Nature Communications]

Reviewers' Comments:

Reviewer #1:

Remarks to the Author:

The authors report enhanced sum-frequency (SF) generation from Au/ZnO nanosponges via the coupling between multiple long-lived plasmonic hot spot fields and excitons in ZnO. In particular, the ultrafast optical dynamics in the hybrid systems have been studied in detail. Although I raised a couple of concerns about the reliability of their data when this manuscript was previously submitted to Nature Nanotechnology. I have checked this revised manuscript and the response letter carefully. I think they almost addressed and resolved the concerns/comments raised by me and other referees, and the data and conclusions are more convincing. Now I believe the femtosecond coherence dynamics and quantum pathways revealed in the coupled systems are novel and should be of interest to the nanophotonics and nonlinear optics communities. The contents also well fit the scope of Nature Communications. Before my recommendation for publication, I would like to suggest the authors to resolve several additional concerns as described below.

1) References could be more comprehensive. For interferometric time-resolved methods to study the dephasing of plasmons, some pioneer works should be cited, e.g., Phys. Rev. Lett., 83, 4421 (1999). For the time-resolved PEEM studies, except for refs 28-30, some other recent works could be cited, e.g., Phys. Rev. B 98, 085128 (2018), Nat. Commun. 9, 4858 (2018), Rev. Sci. Instrum. 90, 113103 (2019) (each work from a different group). Since only few groups can perform the high spatiotemporal resolution time-resolved PEEM studies, comprehensive citations could be fairer.

2) The additive scattering spectra are helpful. However, I'm wondering if the nanosponges used in Figs. 2 and 3 are included in the data of Fig. S3 or not. If so, some discussion about the link between the scattering spectra and the PEEM data/IFRAC data should be provided.

3) I'm still confused why the amplitude of different hot spot fields can be thought to be similar. Here, the dephasing time of different hot spots is of difference (10 -17 fs). I suppose that the field enhancement of different hot spots can also be largely different. It is also surprising that the nonlinear order of the photoemission from the hot spots can be as low as 2.5 - 3.0. For example, the hot spot 1 has the resonance wavelength around 960 nm (1.29 eV). Three photons have the total photon energy of only 3.87 eV. In principle, it's impossible to enable three-photon photoemission in this case, considering the work function of gold (typically larger than 4.5 eV). If possible, please give some more explanations on these issues.

4) Line 91: "(g)" should be changed to "(f)".

Reviewer #3:

Remarks to the Author:

The authors have addressed my original concerns, and I think the work is at the forefront of space-time plasmonic investigation. Therefore, I recommend publication.

Reviewers' Comments (in black) and Our Responses (in dark blue):

Reviewer #1 (Remarks to the Author):

The authors report enhanced sum-frequency (SF) generation from Au/ZnO nanosponges via the coupling between multiple long-lived plasmonic hot spot fields and excitons in ZnO. In particular, the ultrafast optical dynamics in the hybrid systems have been studied in detail. Although I raised a couple of concerns about the reliability of their data when this manuscript was previously submitted to Nature Nanotechnology. I have checked this revised manuscript and the response letter carefully. I think they almost addressed and resolved the concerns/comments raised by me and other referees, and the data and conclusions are more convincing. Now I believe the femtosecond coherence dynamics and quantum pathways revealed in the coupled systems are novel and should be of interest to the nanophotonics and nonlinear optics communities. The contents also well fit the scope of Nature Communications. Before my recommendation for publication, I would like to suggest the authors to resolve several additional concerns as described below.

Reply: We thank the referee for the positive evaluation of our work.

1) References could be more comprehensive. For interferometric time-resolved methods to study the dephasing of plasmons, some pioneer works should be cited, e.g., Phys. Rev. Lett., 83, 4421 (1999). For the time-resolved PEEM studies, except for refs 28-30, some other recent works could be cited, e.g., Phys. Rev. B 98, 085128 (2018), Nat. Commun. 9, 4858 (2018), Rev. Sci. Instrum. 90, 113103 (2019) (each work from a different group). Since only few groups can perform the high spatiotemporal resolution time-resolved PEEM studies, comprehensive citations could be fairer.

Reply: We thank the referee for suggesting the relevant references and fully agree that a more balanced citation is justified. Since we submitted the original version as a Letter, we included only a limited number (no more than 30) of citations. We did not change this list when resubmitting the paper to Nature Communications. We have now added the references mentioned by the Reviewer and some other pioneering work in the revised manuscript.

Changes: a) On p. 2, we have added the following references on the femtosecond dynamics of plasmonic nanostructures (now refs. 16-19): B. Lamprecht et al., Phys. Rev. Lett. 83, 4421 (1999), J. Lehmann et al., Phys. Rev. Lett. 85, 2921 (2000), C. Ropers et al., Phys. Rev. Lett. 94, 113901 (2005).

b) When introducing interferometric frequency-resolved autocorrelation measurements, we cite now as refs. 24-26 the original Ref. 22 (Schmidt, S. et al. Opt. Exp. 18, 25016 (2010)), Ref. 23 (Stibenz, G. & Steinmeyer, G. Opt. Exp. 13, 2617 (2005)) and additionally A. Anderson et al., Nano Letters 10, 2519 (2010).

c) When introducing interferometric time-resolved photoemission electron microscopy on p. 7, we now add as refs. 34-36: A. Klink et al., Phys. Rev. B 98, 085128 (2018), J. Yang et al, Nat. Commun. 9, 4858 (2018) and B. Huber et al., Rev. Sci. Instrum. 90, 113103 (2019).

2) The additive scattering spectra are helpful. However, I'm wondering if the nanosponges used in Figs. 2 and 3 are included in the data of Fig. S3 or not. If so, some discussion about the link between the scattering spectra and the PEEM data/IFRAC data should be provided.

Reply: We agree that a direct comparison of scattering spectra and PEEM/IFRAC data recorded on the same individual nanoparticle could be helpful. Currently, however, we cannot perform such experiments because our home-built dark-field microscope do not provide a fine navigation that allows us to unambiguously locate the specific particles. Therefore, the scattering spectra shown in Fig. S3 have not been recorded on the same nanosponges as those shown in Figs. 2 and 3.

Obviously, the scattering spectra of these particles (see Fig. S3 and J. Zhong et al. Nano Lett. 18, 4957, (2018) and C. Vidal et al., Nano Lett. 18, 1269 (2018)) are very broad with some random variations in spectral intensity that hint some plasmon localization process. These scattering spectra, however, reflect light scattering from an inhomogeneously broadened ensemble of many hot spots and – hence – gives little insight into the plasmonic properties of a single hot spot. The nonlinear IFRAC/PEEM measurements, instead, selectively probe emission from spatially and spectrally confined “hot spot” modes, localized on a 10-nm scale and exhibiting high quality factors and large local field enhancements. As such, it is difficult to directly compare linear and nonlinear scattering spectra – and we would like to refrain from doing this. We emphasize that the near-field scattering spectra that we have recorded earlier (J. Zhong et al., Nano Lett. 18, 4957 (2018)) give direct evidence for the formation of spatially and spectrally confined hot spot modes. Therefore, a comparison of linear and nonlinear light scattering spectra from an individual hot spot would be interesting to create a direct link between linear and nonlinear light scattering / photoemission. Such experiments are currently underway in our laboratory but are beyond the scope of the current manuscript, which focuses on the nonlinear coupling between gold plasmon and ZnO exciton. To confirm that the conclusions that we draw in the present manuscript are indeed statistically meaningful, we have recently analyzed plasmon-enhanced nonlinear emission from a large ensemble of hybrid Au/ZnO nanosponges (J.-M. Yi et al., ACS Photonics 6, 2779 (2019)). These measurements provide independent, additional evidence for the plasmon-enhanced nonlinear exciton emission that strongly supports the conclusions drawn in this manuscript.

Changes: a) When discussing the dark-field white light scattering spectra on page 3, we further write *‘Since the scattering spectra are largely inhomogeneously broadened, they do not give direct insight into the plasmonic properties of single hot spots. Light scattering from individual hot spots has recently been studied using scanning near-field optical spectroscopy, which directly shows that these hot spots are highly localized on a 10-nm scale. The narrow spectral line widths suggest long dephasing times T_2 of ~ 20 fs of hot spots²³’.*

On page 15, we add ‘*Direct evidence for the existence of these hot spot resonances, their long dephasing times and small spatial localization lengths of ~ 10 nm is given by scanning near-field optical spectroscopy and PEEM (Fig. 2) measurements*’.

On page 9, we add ‘*Nonlinear optical spectra taken from many individual ZnO-infiltrated nanosponges confirm this enhancement effect*’.

3) I’m still confused why the amplitude of different hot spot fields can be thought to be similar. Here, the dephasing time of different hot spots is of difference (10 -17 fs). I suppose that the field enhancement of different hot spots can also be largely different. It is also surprising that the nonlinear order of the photoemission from the hot spots can be as low as 2.5 – 3.0. For example, the hot spot 1 has the resonance wavelength around 960 nm (1.29 eV). Three photons have the total photon energy of only 3.87 eV. In principle, it’s impossible to enable three-photon photoemission in this case, considering the work function of gold (typically larger than 4.5 eV). If possible, please give some more explanations on these issues.

Reply: We agree of course that the amplitude of the PEEM signal depends on the field enhancements of the different hot spots and hence can vary from hot spot to hot spot. This is seen, when plotting the interferometric PEEM data on a linear intensity scale (Fig. R1). Experimentally, we typically find for such nanosponges that the intensity of the photoemission signal from different hot spots does not vary too much (Fig. R1). This suggests that the local field enhancement factors of these hot spots do not vary much. This is mainly due to the high order of the nonlinearity in the multi-photon photoemission. This effectively selects out those hot spots with similarly large field enhancements. In Eq. (1), we did not explicitly consider this dependence, because we are most interested in the resonance wavelengths and dephasing times of different hot spots. Since this has caused some confusion, we have now added the amplitude of the localized hot spot mode j , a_j , explicitly in Eq. (1). In the simulation of the IAC traces of the three hot spots, the amplitude of the dipole mode a_{dp} is kept being unity. We have carefully re-checked the simulation and found that the amplitudes of the hot spots 1-3 of 1.1, 0.84 and 0.83, respectively, lead to a good fitting of the experimental data. This indeed shows that the amplitudes of the hot spots do not vary much. We hope that this addresses the concern of the Reviewer.

Figure R1. IAC traces of the data in Fig. 2 plotted in linear intensity scale, showing comparable emission intensity from the three hot spots.

Changes: We have revised Eq. (1) by introducing an amplitude factor a_j of the hot spot modes. We further write in the Methods section, p. 18-19 ‘*For fitting the experimental IAC trace, the amplitude of the dipole mode a_{dp} is kept the same for each simulation of $E_{S,j}$ of the three hot spots. We find that the amplitudes a_j of the hot spots 1-3 are 1.1, 0.84, and 0.83 as large as a_{dp} , respectively.*’

Additionally, the Reviewer is concerned about the order of the nonlinearity, which is slightly lower than the ratio between work function and incident photon energy. Such a lowering of the nonlinear order is quite often observed in multi-photon photoemission from individual metal nanostructures (for some references to our own work, see. C. Ropers et al., Phys. Rev. Lett. 98, 043907, (2007).and G. Hergert et al., Light: Sci. App. 6, e17075, (2017)). Generally, there can be different physical mechanisms that induce such a reduction in nonlinear order. (i) DC bias voltages, applied to the nanostructure, can result in a significant bending of the vacuum level and thus an effective reduction of the work function (C. Ropers et al., Phys. Rev. Lett. 98, 043907, (2007)). In our PEEM, the high DC voltage applied between the sample and the extractor lens can allow electrons to leave the material already at energies lower than the nominal work function. (ii) Possibly, adsorbates on the surface may also contribute to the lowering in nonlinear order. They may lead to additional surface states closer to the vacuum level, from where electrons can be photoemitted. (iii) It is known that photoelectrons are emitted from a hot, nonequilibrium electron system with an effective electron temperature well above room temperature. This effective temperature largely depends on the excitation fluence. Such transient photoelectron heating during multiphoton photoemission and even

deviations from a quasi-thermal distribution have indeed been seen in our earlier work (C. Ropers et al., Phys. Rev. Lett. 98, 043907, (2007)).

Changes: In the Methods section reporting the simulation result, p. 19 we write *‘We note that the observed order of nonlinearity is slightly lower than estimated from a photon picture for the multiphoton-emission of electrons from gold. This may be explained by i) the applied DC bias voltage bending the vacuum level and lowering the work function⁴⁷, ii) emission from local surface (adsorbate) states closer to vacuum level, and iii) emission from a hot, nonequilibrium electron system with an effective electron temperature well above room temperature⁴⁷.’*

4) Line 91: “(g)” should be changed to “(f)”.

Reply: It has been changed.

Reviewer #3 (Remarks to the Author):

The authors have addressed my original concerns, and I think the work is at the forefront of space-time plasmonic investigation. Therefore, I recommend publication.

Reply: We thank the referee for the recommendation of publication of our work.

Reviewers' Comments:

Reviewer #1:

Remarks to the Author:

All the concerns raised in my previous report have been carefully answered and clarified. The revised manuscript is now much more convincing and fits the scope of Nature Communications. Thus, I feel confident to recommend publication in Nature Communications at its present form.